# Annual Vitamin D Status of World-Class British Swimmers Following a Standardised Supplementation Protocol for Three Years

**DOI:** 10.3390/nu17071270

**Published:** 2025-04-05

**Authors:** Josh W. Newbury, Richard J. Chessor, Guy M. Evans, Richard J. Allison, Charlie J. Roberts, Lewis A. Gough

**Affiliations:** 1Research Centre for Life and Sport Science (CLaSS), School of Health Sciences, Birmingham City University, Birmingham B42 2LR, UK; 2Sport Science and Sport Medicine Team, British Swimming, Loughborough, Leicestershire LE11 3QF, UK; 3Circle Health Group, Bath Clinic, Bath BA2 7BR, UK; 4Institute for Sport Exercise and Health (ISEH), University College Hospital London, London NW1 2BU, UK

**Keywords:** micronutrients, swimming, athletes, vitamins

## Abstract

**Background/objectives:** British swimmers are at a heightened risk of vitamin D deficiency (serum 25-hydroxyvitamin D (25(OH)D): <50 nmol∙L^−1^) as their large indoor training volumes often restrict sunlight exposure, especially during the winter when daylight hours are reduced in the United Kingdom. Previous research has recommended supplementation with 4000 IU∙day^−1^ vitamin D_3_ from October to March to offset vitamin D losses. However, no current study has analysed this approach over multiple seasons to assess if this is an appropriate strategy. **Methods:** Using a quasi-experimental design, twenty-nine world-class British swimmers (aged 16–30 years) provided a 10 mL venous blood sample (fasted) as part of their routine haematological screening in the September of three consecutive years (2018, 2019, and 2020). Serum 25(OH)D was determined by radioimmunoassay, and this result determined the length of the standardised vitamin D3 protocol (<30 nmol∙L^−1^: 4000 IU∙day^−1^ from September to March; 30–79 nmol∙L^−1^: 4000 IU∙day^−1^ from October to March; >75 nmol∙L^−1^: no supplementation). **Results:** Mean serum 25(OH)D concentrations increased each year (2018: 76.4 ± 28.4 nmol∙L^−1^, 2019: 91.5 ± 24.8 nmol∙L^−1^, 2020: 115.0 ± 36.6 nmol∙L^−1^, *p* < 0.001), which coincided with the eradication of vitamin D deficiency after one season (prevalence, 2018: 10%, 2019: 0%, 2020: 0%). In September 2020, 35% of swimmers had a serum 25(OH)D > 125 nmol∙L^−1^, although it is currently debated whether this is a concern or a benefit for athletic populations. **Conclusions:** Supplementing with 4000 IU∙day^−1^ of vitamin D_3_ throughout the winter can increase the vitamin D status of swimmers. However, more frequent testing may be required to ensure that serum 25(OH)D remains within the sufficient range across the season (75–125 nmol∙L^−1^).

## 1. Introduction

The past decade has overseen an emergence of evidence detailing the importance of vitamin D for optimal physiological function, including essential roles in immunity, skeletal muscle remodelling, protein synthesis, and bone health [1,2,3]. With each of these functions also associated with long-term physical performance, it is unsurprising that athletes are encouraged to maintain a sufficient vitamin D status (serum 25-hydroxyvitamin D (25(OH)D) concentration ≥ 75 nmol∙L^−1^) year-round to support training adaptations and reduce injury risk [3,4]. Approximately 80–90% of 25(OH)D is synthesised through a complex reaction involving ultraviolet-B radiation (e.g., sunlight) and the skin [3], therefore maintaining a sufficient status can be problematic in the autumn and winter due to poor weather conditions and diminishing hours of daylight in the United Kingdom [2,5]. Furthermore, athletes that complete a large quantity of indoor training (e.g., swimmers) are at a greater risk of insufficient vitamin D (serum 25(OH)D < 75 nmol∙L^−1^) as direct sun exposure can easily be avoided throughout the year [4,6]. Indeed, studies have identified a high prevalence of vitamin D insufficiency in swimmers from Denmark (45% [7]) and Virginia, USA (79% [8]), whereas the mean 25(OH)D of swimmers from the warm weather climates of Israel were still unable to reach the sufficient threshold (56.9–69.3 nmol∙L^−1^ [9,10,11]). A 13% decline in serum 25(OH)D concentrations has been observed in high-level adolescent British swimmers from October to January [12]. Considering that British swimmers cannot solely rely on sunlight exposure to obtain a sufficient vitamin D status, it is clear that this population could benefit from increasing their dietary vitamin D intake.

A dietary intake of 400 IU∙day^−1^ (10 µg∙day^−1^) is currently recommended to avoid vitamin D deficiency (serum 25(OH)D < 50 nmol∙L^−1^) in Great Britain [13,14]. This is equivalent to consuming 70 g of herring, 85 g of salmon, or 5 large eggs (250 g) every day [15], which a target that is often not achieved in the diets of swimmers (84–368 IU∙day^−1^ [7,16,17,18]. Combined with a lack of sunlight exposure, a failure to consume adequate vitamin D has been shown to cause 30 nmol∙L^−1^ decrements in 25(OH)D within the first 3–4 months of the autumn/winter swimming season (August to November/December) [8,19]. Over the same timeframe, however, supplementation with 4000 IU∙day^−1^ or 5000 IU∙day^−1^ of vitamin D3 increased 25(OH)D concentrations by +20 nmol∙L^−1^ (August to December [19]) and +14 nmol∙L^−1^ (August to November [8]), respectively. Interestingly, Lewis et al. [19] continued supplementation for a further three months (August to March), yet this extended time did not continuously increase 25(OH)D concentrations (March vs. August: +3 nmol∙L^−1^). Based on this evidence, indoor athletes are encouraged to supplement with 4000 IU∙day^−1^ of vitamin D3 in the autumn and winter in order to sustain a sufficient vitamin D status throughout the entire competitive season [20]. Despite this suggestion, no research to date has investigated the cumulative effect that vitamin D3 might have on the annual vitamin D status of swimming athletes. This study therefore retrospectively analysed the three-year change in 25(OH)D that was collected from world-class British swimmers following their annual ingestion of 4000 IU∙day^−1^ of vitamin D3. The hypothesis of this study was that supplementation with vitamin D3 will improve the annual concentrations of vitamin D over the observation period.

## 2. Materials and Methods

### 2.1. Experimental Overview

This study was a quasi-experimental design with observational and interventional components. The observation component consisted of three experimental visits that were part of a general health screening for world-class British swimmers to determine serum 25-hydroxyvitamin D (25(OH)D) concentrations. Visits were carried out annually every September between 2018 and 2020. The interventional component was that participants are encouraged to ingest vitamin D as per the guidance of the British Swimming’s Chief Medical Officer and experienced sport nutritionists in collaboration with the English Institute of Sport (now UK Sports Institute). Once informed of the risks of being involved, including the risks of vitamin D ingestion, all swimmers gave their informed consent to have their 25(OH)D data anonymised for the purpose of this study. This study was granted ethical approval by Birmingham City University (Newbury/#10152/sub2/R(A)/2019/Jan/HELS FAEC).

### 2.2. Participants

Twenty-nine world-class British swimmers (aged 16–30 years) were involved in this study. Each participant was a nationally funded athlete that was competing at the international level between 2018 and 2020 (mean age, 2018: 21.0 ± 3.0 years; 2019: 22.0 ± 3.0 years; 2020: 23.0 ± 3.0 years) and aiming for selection for Olympic events. All swimmers trained in indoor venues in Great Britain, which are all located at latitudes > 51° N. Swimmers typically trained twice per day between six and seven days per week and consisted of pool swimming and strength and conditioning land-based sessions. In September of each year, all swimmers provided a venous blood sample that was analysed for serum 25(OH)D as part of a wider haematological screen.

### 2.3. Blood Sampling Procedure

All swimmers attended British Swimming’s medical facilities and fasted to give a 10 mL blood sample from the antecubital vein of the forearm. Blood was drawn by a clinically trained phlebotomist and stored at room temperature for <60 min until all selected swimmers had given their sample for that day. Once blood collection was complete, the samples were immediately couriered to an independent laboratory (2018 and 2019: Queen’s Medical Centre, Nottingham; 2020: Nationwide Pathology, Lutterworth, Leicestershire) for radioimmunoassay analysis. For the purpose of this study, serum 25(OH)D concentrations were interpreted as ‘severely deficient’: <25 nmol∙L^−1^, ‘deficient’: 25–49 nmol∙L^−1^, ‘insufficient’: 50–74 nmol∙L^−1^, ‘sufficient’: 75–125 nmol∙L^−1^, ‘high, but reportedly safe’: 126–250 nmol∙L^−1^, and ‘potentially toxic’: >250 nmol∙L^−1^ in accordance with previous research [3,21,22].

### 2.4. Supplementation Protocol

All swimmers were provided with a sufficient amount of vitamin D3 capsules (4000 IU, Elite Vitamin D3, Healthspan, Guernsey, UK) to last the entire dosing period (Table 1) and were instructed to request a repeat provision should their supply run out. Alongside verbal communication, a short advice sheet was provided to the swimmers to improve their understanding of vitamin D3 supplementation. No prescriptive instructions were given on when to take the supplement, but advice was given to take it at a time when the individual was most likely to keep consistent (e.g., with breakfast or last thing at night). No further guidance was given regarding other sources of either dietary or supplemental vitamin D; therefore, most swimmers may have also consumed an additional 200 IU∙day^−1^ through a daily multivitamin (Elite Gold A–Z, Healthspan, Guernsey, UK). Although each swimmer was allocated the same amount of daily vitamin D3, the advised timeframes for supplementation differed per individual based upon the results of their September 25(OH)D screening results. Supplement protocols were selected and managed by British Swimming’s Chief Medical Officer and experienced sport nutritionists in collaboration with the English Institute of Sport. The protocols coincide with recommendations from previous research in swimmers [19] and are within the safe upper limit for adults and adolescents [13]. Throughout the study, there were no systematic attempts to monitor supplement adherence.

### 2.5. Statistical Analysis

Data were normally distributed (Shapiro–Wilks) and sphericity was not violated (Mauchly) prior to statistical analysis. A repeated measures ANOVA was performed to establish mean differences between the serum 25(OH)D concentrations at each time point (2018, 2019, 2020). Post hoc comparisons were determined by the Bonferroni correction, and main effect sizes are reported as partial eta squared (Pη2). Main effect sizes were interpreted as ‘small’ (0.01–0.05), ‘medium’ (0.06–0.13), and ‘large’ (>0.13) [23]. In addition, Cohen’s d effect sizes were calculated for yearly 25(OH)D comparisons (2018 vs. 2019, 2019 vs. 2020, and 2018 vs. 2020), which were interpreted as ‘trivial’ (<0.20), ‘small’ (0.20–0.49), ‘medium’ (0.50–0.79), and ‘large’ (≥0.80) [23]. The coefficient of variation (CV) was calculated for the inter-individual differences in 25(OH)D change over time using SD/mean x 100. All statistical tests were completed using the Statistical Package for the Social Sciences (SPSS), version 25 (IBM, Chicago, IL, USA). All data are reported as mean ± SD with statistical significance set at *p* ≤ 0.05.

## 3. Results

Mean 25(OH)D concentrations increased across the sampling timeframe (*p* < 0.001, pŋ2 = 0.47), with these changes occurring in an incremental fashion from 2018 (76.4 ± 28.4 nmol∙L^−1^) to 2019 (91.5 ± 24.8 nmol∙L^−1^, *p* = 0.035, d = 0.57), and again from 2019 to 2020 (115.0 ± 36.6 nmol∙L^−1^, *p* = 0.001, d = 0.75; Figure 1). Over the two-year supplement period, there was a 50.5% increase in the mean serum 25(OH)D concentrations of world-class British swimmers (+ 38.6 nmol∙L^−1^, *p* < 0.001, d = 1.18).

Seven percent (2 of 29) of swimmers had a deficient vitamin D status in 2018, including one individual that was classified as severely deficient (23 nmol∙L^−1^). This was the only year that vitamin D deficiency was observed across the sampling timeframe with all remaining 25(OH)D concentrations ≥ 50 nmol∙L^−1^ in 2019 and 2020 (Table 2). Insufficient 25(OH)D concentrations more than halved on a yearly basis, with 55% (*n* = 19) identified in 2018, 31% (*n* = 9) in 2019, and 7% (*n* = 2) in 2020. Similarly, the highest recorded 25(OH)D concentration increased each year from 136 nmol∙L^−1^ in 2018 to 163 nmol∙L^−1^ in 2019, and finally 193 nmol∙L^−1^ in 2020.

## 4. Discussion

This was the first study to have monitored the annual change in serum 25(OH)D concentrations in world-class British swimmers. In turn, this is valuable to practitioners who may use these data as normative. Moreover, this study investigated how this cohort responded to an individualised 4000 IU∙day^−1^ vitamin D3 protocol. Within one year of supplementation, vitamin D deficiency (<50 nmol∙L^−1^) was completely eradicated from this cohort. Moreover, mean serum 25(OH)D increased on a yearly basis until almost all swimmers (93%) displayed a sufficient vitamin D status (>75 nmol∙L^−1^) by the 2020 timepoint. Whilst not all these changes can be attributed to supplementation alone, the data suggest that, in part, supplementation could be an option for practitioners that is successful and safe.

In September 2018, world-class British swimmers had a group mean 25(OH)D of 76 ± 28 nmol∙L^−1,^ suggesting that the majority of this cohort had sufficient vitamin D concentrations [3,6]. However, based on previous studies, an autumn 25(OH)D concentration above 123 nmol∙L^−1^ may be necessary to sustain serum 25(OH)D > 75 nmol∙L^−1^ throughout the winter months [24]. Indeed, a multi-cultural cohort of footballers based in Great Britain failed to maintain their ‘sufficient’ status (August: 104 ± 21 nmol∙L^−1^) even after 18 weeks of daily outdoor training (December: 51 ± 19 nmol∙L^−1^ [25]). This is more concerning for swimmers when analysing individual data, which identified that September 25(OH)D levels were below 75 nmol∙L^−1^ in two-thirds of the current cohort. Given that this measurement was taken immediately following Britain’s peak sunlight hours (May to August), this would suggest that British swimmers could be at a high risk of deficiency during the winter without vitamin D3 supplementation. Without a winter measurement, it cannot be confirmed based on this study how 4000 IU∙day^−1^ affected the serum 25(OH)D concentrations of world-class swimmers. Nonetheless, based on previous research, NCAA Division I swimmers maintained mean concentrations of 100 nmol∙L^−1^ from October to March when using the same supplemental dose [18]. This would partly explain the incremental increases in serum 25(OH)D that were observed each year. We also note the inter-individual variation within the current study. As limited compliance checks were employed in the current study, this may explain why such variation occurred, and therefore future studies should consider pill counting, diaries or digital monitoring. It can be inferred that compliance was good from most of the current study sample, however, as the concentrations generally improved year on year. Finally, future studies are therefore required with increased sampling frequencies (i.e., September, December, April) to assess whether supplementation protocols ensure sufficient vitamin D concentrations across the entire season.

The increases in group mean 25(OH)D that occurred each year resulted in almost all participants (93%) achieving a ‘sufficient’ vitamin D status for the beginning of the 2020/21 season. Though this outcome is seemingly beneficial, a controversial finding was that ‘high’ 25(OH)D concentrations became more commonplace (>125 nmol∙L^−1^; 2018: 10%, 2020: 35%). Based on the Endocrine Society guidelines, there are no risks of toxicity (i.e., hypercalcemia) when 25(OH)D is <375 nmol∙L^−1^ [26]. To ensure that this concentration is not exceeded, a safety threshold of <250 nmol∙L^−1^ is therefore recommended [26], which was not surpassed by any swimmer in this study (peak recorded 25(OH)D: 193 nmol∙L^−1^). Conversely, the UK National Health Service (NHS) uses more conservative guidelines akin to the American Dietetic Association (ADA), whereby serum 25(OH)D concentrations ≥ 150 nmol∙L^−1^ are associated with possible adverse effects [13]. These alternative guidelines could place 24% of current swimmers at risk of symptoms, such as nausea, dehydration, and lethargy [27,28]; however, this is uncertain since more intense dosing protocols (i.e., 10,000 IU∙day^−1^ for 20 weeks) have previously been well tolerated in healthy adults [29,30]. One explanation is that excessive vitamin D exposure increases the rate of vitamin D catabolism, subsequently creating a negative feedback loop that tightly regulates 25(OH)D < 250 nmol∙L^−1^ [3,31]. However, this study cannot support this notion since no side effects of vitamin D toxicity were measured. Nonetheless, with no additional benefits to exceeding the ‘sufficient’ threshold [3], the current study approach halted supplementation when serum 25(OH)D exceeded 125 nmol∙L^−1^ and it is therefore suggested this is adopted with those with higher concentrations at baseline/first assessment. It is therefore speculated that serum 25(OH)D levels would decline in ‘at risk’ swimmers, though this requires a follow-up investigation.

At present, there are no serum 25(OH)D guidelines that are considered optimal for world-class athletes. Some suggestions state that athletes should maintain a 25(OH)D ≥ 100 nmol∙L^−1^ to ensure that adequate amounts are available to support musculoskeletal health [4]. This is due to the high prevalence of vitamin D insufficiency in elite cohorts (30%) that is typically higher than in the general population [32]. Moreover, sustaining serum 25(OH)D > 120 nmol∙L^−1^ is associated with less frequent and severe upper respiratory tract infections (URTI) in endurance athletes during the winter [33]. Unpublished data from within this group found that URTIs accounted for 50% of missed training sessions in the 2016–2017 swimming season, with 56% of the world-class cohort experiencing at least one URTI instance. Fewer URTI cases could increase the consistency of training intensity, subsequently enabling greater training-induced adaptations [34]. As the prevalence of URTIs was not measured alongside the other measures in the study, future work should investigate this to ascertain if a link exists. Conversely, no direct ergogenic effects are expected with vitamin D3 supplementation unless it involves the correction of neuromuscular defects caused by deficiency [2,3,35,36]. This has recently been contested, however, since 5000 IU∙day^−1^ vitamin D3 improved deadlift (vitamin D: +13.8%, placebo: +2.5%) and vertical jump (vitamin D: +13.5%, placebo: +2.1%) performance in collegiate swimmers [8]. Interestingly, both groups started (vitamin D: 112.1 ± 8.2 nmol∙L^−1^, placebo: 117.0 ± 15.4 nmol∙L^−1^) and ended the 12-week supplement protocol with a sufficient vitamin D status (vitamin D: 131.3 ± 28.6 nmol∙L^−1^, placebo: 81.3 ± 17.0 nmol∙L^−1^). However, the group that supplemented with vitamin D3 displayed a better maintenance of free testosterone levels and sex-hormone binding globulin over the research timeframe. Though unclear whether this would directly improve swimming performance, further research is warranted to determine whether a 25(OH)D > 120 nmol∙L^−1^ could support training adaptations whilst also maintaining health status.

Serum 25(OH)D is an appropriate marker of vitamin D status based on its long half-life and close association with vitamin D exposure (e.g., sunlight, food, supplementation) [3]. Whilst this measure has been discussed with relation to possible effects on illness and performance, a limitation is that other vitamin D metabolites (i.e., free 25(OH)D) have greater responsibilities within bone health [37,38,39]. Swimmers are consistently reported to have lower bone mineral density (BMD) compared to other athletes due to the non-weight-bearing nature of their sport [40,41,42]; therefore, monitoring free 25(OH)D alongside serum 25(OH)D may be an important addition to the health screening process. Groups of ethnically diverse swimmers also warrant the measurement of multiple vitamin D metabolites since differences exist in skin pigmentation (i.e., ultraviolet-B exposure) and genetic polymorphisms (i.e., vitamin D binding protein, vitamin D receptor phenotypes) that may be at greater risk of deficiency [43,44,45]. Due to these limitations, world-class swimming programmes are encouraged to screen multiple vitamin D markers to accurately determine the ‘true’ vitamin D status of their athletes. Another limitation was that serum 25(OH)D was measured in September of each year; therefore, whether this protocol offset deficiency over the winter months remains unclear. Future research should ensure that a sample during the winter is taken to capture the seasonal effects on vitamin D concentrations. This would also help, as with high doses of vitamin D over a considerable time frame (>6 months), markers for routine bone markers would also be appropriate to monitor. Equally, daily intake of calcium should be monitored, as a high intake of vitamin D in the presence of a potentially limited calcium intake could lead to accrued bone resorption. Finally, future research should consider using a control group to make greater inferences on the effects of supplementation.

## 5. Conclusions

This study reports the three-year concentrations in world-class British swimmers, which has never been explored before. The findings suggest that intervention via sports nutritionists, at least in part, eradicated vitamin D deficiency within one calendar year and helped maintain the eradication over a three-year period. Based on the study findings, vitamin D supplementation should be used with those that are deficient only. These consistent increases in serum 25(OH)D also led to 35% of swimmers developing ‘high’ concentrations (>125 nmol∙L^−1^), which raises concerns regarding toxicity when this supplement strategy is used for more than two years. Despite no health issues being observed, it may be intuitive to suggest that those sufficient may require a lower dose of vitamin D. Due to the individual variation in responses, testing and maintenance are key, and future research should also include additional compliance measures. Further research is therefore required to determine whether there is an optimal serum 25(OH)D range that benefits the immunity and physical performance of highly trained athletes.

## Figures and Tables

**Figure 1 nutrients-17-01270-f001:**
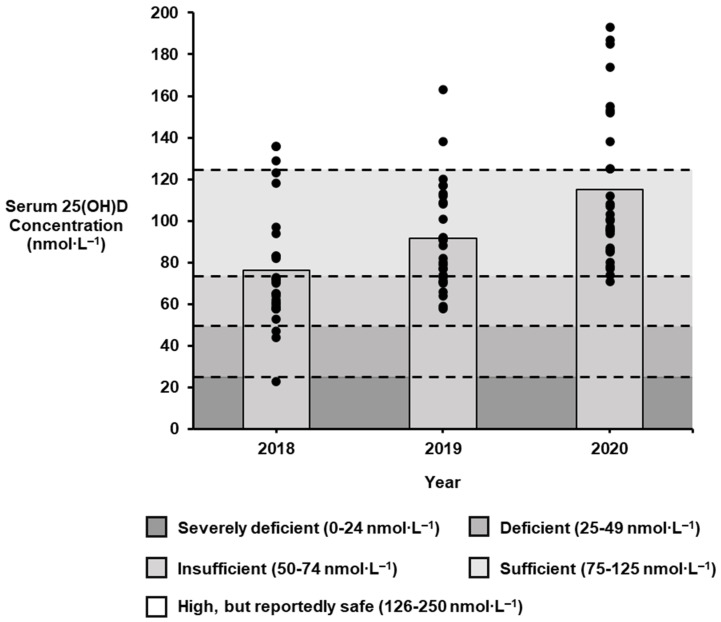
Yearly change (2018–2020) in the serum 25-hydroxyvitamin D (25(OH)D) concentrations of world-class British swimmers. ● signifies individual data points.

**Table 1 nutrients-17-01270-t001:** Individualised vitamin D_3_ supplementation protocols assigned to swimmers as determined by baseline serum 25-hydroxyvitamin-D (25(OH)D) concentration.

Serum 25 (OH) D Concentration	Supplementation	Timeframe	Further Recommendation
<30 nmol∙L^−1^	4000 IU∙day^−1^	Immediately (September–March)	Retest 25(OH)D after six weeks, including parathyroid hormone and alkaline phosphatase
30–50 nmol∙L^−1^	4000 IU∙day^−1^	Immediately (September–March)	Retest 25(OH)D after six weeks
51–80 nmol∙L^−1^	4000 IU∙day^−1^	24 weeks (October–March)	Safely increase sunlight exposure in the summer months
81–125 nmol∙L^−1^	4000 IU∙day^−1^	24 weeks (October–March)	
>125 nmol∙L^−1^	No supplementation		

**Table 2 nutrients-17-01270-t002:** Elite British swimmers’ vitamin D status between 2018 and 2020.

Serum 25(OH)D Concentration (nmol∙L^−1^)	Swimmers in Each Category (%)		
	**2018**	**2019**	**2020**
0–24 *‘severely deficient’*	3.4	0	0
25–49 *‘deficient’*	6.9	0	0
50–74 *‘insufficient’*	55.2	31.0	6.9
75–125 *‘sufficient’*	27.5	62.1	58.6
126–250 *‘high, but reportedly safe’*	6.9	6.9	34.5

## Data Availability

The datasets generated during and/or analysed during the current study are available from the corresponding author on reasonable request.

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
