# Peer review of "Annual Vitamin D Status of World-Class British Swimmers Following a Standardised Supplementation Protocol for Three Years"

_nutrients, 2025, doi:10.3390/nu17071270_

Round 1

Reviewer 1 Report (Previous Reviewer 2)

Comments and Suggestions for Authors

The authors improved the text to the possible extent.

Comments on the Quality of English Language

Good

Author Response

Author response to comments

Reviewer comment: In blood sampling procedure, we suggest stipulating if it was in fasting state.

Response: We have now added this

Reviewer comment: The recommendation as 400 IU by day, are probably not adequate for elite athletes, and apparently good sun exposure do not resolve? It should be of interest to discuss what should be an appropriate consumption for elites’ sports having sufficient serum 25(OH)D level? Or it is necessary mega consumption in this population like 4000 IU that is 10 time the general population recommendation?

Response: We have now placed this into the manuscript on lines 224 and 284.

Reviewer comment: About the compliance, we understand that you have not controlled, but probably you have some indication of the swimmers? By the results obtained, we think that the compliance was generally good.

Response: We have added this on lines 224 and 200

Reviewer 2 Report (Previous Reviewer 3)

Comments and Suggestions for Authors

Dear authors, I see that the current version of the text is of sufficient quality to be published by Nutrients. In the previous version, I had already approved the text and I see that the authors have made efforts to further improve the text to satisfy the other reviewers. In this sense, I maintain my previous decision, which is to recommend publication again.

Author Response

Author response to comments

Reviewer comment: In blood sampling procedure, we suggest stipulating if it was in fasting state.

Response: We have now added this

Reviewer comment: The recommendation as 400 IU by day, are probably not adequate for elite athletes, and apparently good sun exposure do not resolve? It should be of interest to discuss what should be an appropriate consumption for elites’ sports having sufficient serum 25(OH)D level? Or it is necessary mega consumption in this population like 4000 IU that is 10 time the general population recommendation?

Response: We have now placed this into the manuscript on lines 224 and 284.

Reviewer comment: About the compliance, we understand that you have not controlled, but probably you have some indication of the swimmers? By the results obtained, we think that the compliance was generally good.

Response: We have added this on lines 224 and 200

Reviewer 3 Report (New Reviewer)

Comments and Suggestions for Authors

Dear Authors,

This is a nice article, that for sure will be of interest for all stakeholders around elite swimmers’ preparation, but also for other sports modalities. We have no concerns on your study, just some suggestions that we hope are susceptible to improve some parts.

In blood sampling procedure, we suggest to stipulate if it was in fasting state.

The recommendation as 400 IU by day, are probably not adequate for elite athletes, and apparently good sun exposure do not resolve? It should be of interest to discuss what should be an appropriate consumption for elites’ sports having sufficient serum 25(OH)D level? Or it is necessary mega consumption in this population like 4000 IU that is 10 time the general population recommendation?

About the compliance, we understand that you have not controlled, but probably you have some indication of the swimmers? By the results obtained, we think that the compliance was generally good.

Wishes of success

Author Response

Author response to comments

Reviewer comment: In blood sampling procedure, we suggest stipulating if it was in fasting state.

Response: We have now added this

Reviewer comment: The recommendation as 400 IU by day, are probably not adequate for elite athletes, and apparently good sun exposure do not resolve? It should be of interest to discuss what should be an appropriate consumption for elites’ sports having sufficient serum 25(OH)D level? Or it is necessary mega consumption in this population like 4000 IU that is 10 time the general population recommendation?

Response: We have now placed this into the manuscript on lines 224 and 284.

Reviewer comment: About the compliance, we understand that you have not controlled, but probably you have some indication of the swimmers? By the results obtained, we think that the compliance was generally good.

Response: We have added this on lines 224 and 200

This manuscript is a resubmission of an earlier submission. The following is a list of the peer review reports and author responses from that submission.

Round 1

Reviewer 1 Report

Comments and Suggestions for Authors

This work examines vitamin D status of supplemented professional swimmers during a period of three years. The problem I see in this study is the lack of measurements and variables, not only that 25(OH)D levels were determined once per year, but also there is not data regarding other factors that may influence vitamin D status (e.g. diet or sunlight exposure). 25(OH)D levels were measured in September, after summer, when sun exposure usually is much higher. Additionally, not frequency of supplements consumption was recorded…Thus, it is impossible to know certainly if the improvement in vitamin D status was caused by the intervention. Maybe, the only valid conclusion is that the intervention protocol seems to be safe.

Author Response

Response to reviewer comments

The authors would like to thank the reviewers for their hard work and comments to improve our manuscript, below we include a point-by-point response.

This is an important manuscript about vitamin D supplementation in swimmers. I will present below my considerations about the text, point-by-point.

I would like to suggest changes to the title, I believe that a more direct title might attract more reader interest. I suggest something like: Vitamin D supplementation improves status in winter swimmers. Please review.

Author response: We acknowledge the suggestion of a more direct title, but given the limitations of the study we would argue to keep it as is. This is because we only suggest supplementation is causal and we haven’t measured enough other variables to allow us to be so direct.

The Abstract is adequately structured. The introduction is clearly concise and has the appropriate elements, except the hypothesis, please add the hypothesis at the end of the introduction.

Author response: This has now been included.

The methods are also well described, but I ask the authors to create a subhead at the beginning of the methods: 2.1. Experimental approach, where authors must present a general description of the experimental protocol and present the ethical aspects.

Author response: This has now been added under heading 2.1

Enter the number of hours trained by the athletes per week and the competitive level of the athletes into the methods.

Author response: We have now added this on line 92. We have not provided the number of hours trained as unfortunately we do not have access to this data.

Why did the authors choose to supplement athletes who had between 81-125 nmol∙L− 1 considering that it is a range in which they consider that athletes already have sufficient levels of vitamin D.

Author response: This was not the decision of the authors. These protocols are decided by British Swimming and we had not input on this process.

Table 2 does not need a legend, the categories can be described alongside the concentrations.

Author response: We have now done this, thanks for the suggestion.

Given the effects of vitamin D on immunity, has there been any monitoring of a lower rate of infection in athletes? It would be interesting if the study tried to only measure biochemical indicators and there were also practical applications of the effect of supplementation. Furthermore, it is important that authors give voice to authors who argue against the need for vitamin D supplementation.

Author response: Unfortunately, we did not measure anything that would provide a practical insight. Our findings are more to show other practitioners what a high-performance training center does and what success (or lack) of a support protocol has.

https://link.springer.com/article/10.1007/s40279-017-0841-9

https://link.springer.com/article/10.1007/s40279-017-0841-9

Therefore, I think it is important that the authors also analyze whether there are symptoms of diseases caused by insufficient vitamin D. After all, if the athletes did not have any health problems, would it really be necessary to supplement vitamin D? I would like to suggest that the authors include reflections on what I suggested above in the discussion.

Author response: We have now included this to pose the question. This can be seen in line 230-235.

Reviewer 2

This work examines vitamin D status of supplemented professional swimmers during a period of three years. The problem I see in this study is the lack of measurements and variables, not only that 25(OH)D levels were determined once per year, but also there is not data regarding other factors that may influence vitamin D status (e.g. diet or sunlight exposure). 25(OH)D levels were measured in September, after summer, when sun exposure usually is much higher. Additionally, not frequency of supplements consumption was recorded…Thus, it is impossible to know certainly if the improvement in vitamin D status was caused by the intervention. Maybe, the only valid conclusion is that the intervention protocol seems to be safe.

Author response: We have toned down the discussion and conclusion and in various other areas of the manuscript to acknowledge this. We have mainly done this via stating that the intervention was “in part” the cause for improvements in concentration of Vitamin D. We have also added a note in the conclusion that the approach seems to be safe.

Reviewer 3

This article addresses a timely and practically relevant question: whether annual supplementation with 4000 IU/day of vitamin D3 can sustain or improve vitamin D status in elite British swimmers across multiple seasons. The longitudinal design is a particular strength, as few studies examine changes over numerous years. By sampling blood in the late summer/early fall (when vitamin D levels are typically at their annual peak in the Northern Hemisphere) and tracking year-to-year changes, the authors provide an important snapshot of how such a supplementation protocol influences vitamin D status over an extended timeframe.

Below is a more in-depth critique assessing the study’s objectives, methodology, results, and interpretation.

  1. Blood samples were only collected each September. While this measures vitamin D status when levels are typically highest at a time of year, it limits insight into whether the protocol effectively maintains sufficiency during the winter and spring—precisely when deficiency is more common. Additional sampling in mid-winter or quarterly terms would strengthen the evidence that the 4000 IU/day regimen prevents drops below recommended thresholds.

Author response: We certainly acknowledge this and have added a section on this topic in line 247-250.

  1. Although the total cohort of 29 is not unusually small for an elite athlete population, it still limits the ability to generalize findings to other sports or the broader population. Indeed, world-class athletes may differ significantly from amateur swimmers or less specialized groups. Additionally, the description lacks a more precise account of international-level athletes. Including athletes’ personal best results or their success on the international stage could enhance the analysis and be beneficial.

Author response: We acknowledge these findings cannot be generalized to other sports. We do suggest that the sample size we have is large based on the pool of swimmers at this level available. These athletes were funded international athletes so aiming towards Olympic success. We have edited this section slightly on line 78.

  1. While swimmers received adequate vitamin D3 capsules and some basic education, no systematic measure of actual adherence (e.g., pill counts, diaries, or digital monitoring) was reported. Non-adherence may explain the inter-individual variability in 25(OH)D levels.

Author response: This is an excellent suggestion and we have now added this to line 181-182. The compliance checks in this study were verbal as the nutrition staff are on the ground with the swimmers.

  1. The authors state that swimmers may have consumed small amounts of vitamin D from a daily multivitamin and that no specific guidance was given on dietary intake. Sun exposure differences, travel to warm-weather training camps, and nutritional behaviors can significantly affect 25(OH)D levels, and these confounders are not rigorously accounted for.

Author response: We acknowledge this point. We have reworded the beginning of the discussion and conclusion to assist with making the point above.

  1. Measurement of serum 25(OH)D alone is standard and appropriate, but as the authors note, additional markers like free 25(OH)D, parathyroid hormone (PTH), or bone turnover markers could yield deeper insights. Monitoring these would be beneficial for a fuller picture of bone health, especially in swimmers, who are prone to lower bone mineral density.

Author response: We do acknowledge this and we have state this on line 239.

  1. A proportion of swimmers (>35%) exceeded 125 nmol∙L^−1. While the article rightly discusses safety cutoffs (variously 150 nmol∙L^−1 vs. 250 nmol∙L^−1) and potential hypercalcemia risk, evidence of adverse outcomes below 250 nmol∙L^−1 is still mixed. Thus, the practical significance of “high” 25(OH)D remains controversial, especially in elite athletes who may benefit from robust vitamin D status for immunity and musculoskeletal health.

Author response: We agree with the reviewer here. We wanted to provide some balance to this argument, given that these narratives exist.

  1. While the study mentions ethical approval, there is limited detail on whether participants were fully informed of potential risks associated with high-dose vitamin D supplementation.

Author response:

  1. The reviewer also noted that several words are awkwardly hyphenated or split across lines, likely due to a formatting or typesetting artifact. Examples include: 
  2. “tes- tosterone” (should be “testosterone”)
  3. “rec- ommendations” (should be “recommendations”)
  4. “expo- sure” (should be “exposure”)
  5. “ad- equate” (should be “adequate”)
  6. “in- vestigation” (should be “investigation”)
    Author response: Unfortunately, this is due to the type setting of the Nutrients template so we cannot correct this.

Final remarks:

  1. As a reviewer, I recommend that the authors revise the “Conclusions” to emphasize their findings' practical and clinical implications without restating specific results. Please focus on how 4000 IU/day vitamin D₃ supplementation reduces deficiency and supports sufficient vitamin D levels in elite swimmers. Highlight the role of individualized dosing and suggest more frequent (especially winter) testing to optimize and avoid excessive levels. This succinct approach will give clear, actionable guidance for practitioners and avoid duplication of the detailed data reported earlier.

Author response: This has now been completed and most of it has been rewritten

  1. I recommend including a “Limitations section” to strengthen the manuscript's scientific rigor. Key aspects to address include the lack of a control group, which restricts causal interpretation, and the seasonal timing of measurements, which may not fully capture fluctuations in vitamin D levels. Additionally, the study's focus on elite British swimmers’ limits generalizability, and potential confounders, such as dietary intake and sun exposure, were not controlled. Clarifying these factors will provide a more balanced interpretation of the findings and ensure the manuscript aligns with standard reporting practices.

Author response: We acknowledge these issues, and our limitations are currently on Lines 236-254.

Reviewer 2 Report

Comments and Suggestions for Authors

Overall Impression

This article addresses a timely and practically relevant question: whether annual supplementation with 4000 IU/day of vitamin D3 can sustain or improve vitamin D status in elite British swimmers across multiple seasons. The longitudinal design is a particular strength, as few studies examine changes over numerous years. By sampling blood in the late summer/early fall (when vitamin D levels are typically at their annual peak in the Northern Hemisphere) and tracking year-to-year changes, the authors provide an important snapshot of how such a supplementation protocol influences vitamin D status over an extended timeframe.

Below is a more in-depth critique assessing the study’s objectives, methodology, results, and interpretation.

  1. Blood samples were only collected each September. While this measures vitamin D status when levels are typically highest at a time of year, it limits insight into whether the protocol effectively maintains sufficiency during the winter and spring—precisely when deficiency is more common. Additional sampling in mid-winter or quarterly terms would strengthen the evidence that the 4000 IU/day regimen prevents drops below recommended thresholds.
  2. Although the total cohort of 29 is not unusually small for an elite athlete population, it still limits the ability to generalize findings to other sports or the broader population. Indeed, world-class athletes may differ significantly from amateur swimmers or less specialized groups. Additionally, the description lacks a more precise account of international-level athletes. Including athletes’ personal best results or their success on the international stage could enhance the analysis and be beneficial.
  3. While swimmers received adequate vitamin D3 capsules and some basic education, no systematic measure of actual adherence (e.g., pill counts, diaries, or digital monitoring) was reported. Non-adherence may explain the inter-individual variability in 25(OH)D levels.
  4. The authors state that swimmers may have consumed small amounts of vitamin D from a daily multivitamin and that no specific guidance was given on dietary intake. Sun exposure differences, travel to warm-weather training camps, and nutritional behaviors can significantly affect 25(OH)D levels, and these confounders are not rigorously accounted for.
  5. Measurement of serum 25(OH)D alone is standard and appropriate, but as the authors note, additional markers like free 25(OH)D, parathyroid hormone (PTH), or bone turnover markers could yield deeper insights. Monitoring these would be beneficial for a fuller picture of bone health, especially in swimmers, who are prone to lower bone mineral density.
  6. A proportion of swimmers (>35%) exceeded 125 nmol∙L^−1. While the article rightly discusses safety cutoffs (variously 150 nmol∙L^−1 vs. 250 nmol∙L^−1) and potential hypercalcemia risk, evidence of adverse outcomes below 250 nmol∙L^−1 is still mixed. Thus, the practical significance of “high” 25(OH)D remains controversial, especially in elite athletes who may benefit from robust vitamin D status for immunity and musculoskeletal health.
  7. While the study mentions ethical approval, there is limited detail on whether participants were fully informed of potential risks associated with high-dose vitamin D supplementation.
  1. The reviewer also noted that several words are awkwardly hyphenated or split across lines, likely due to a formatting or typesetting artifact. Examples include: 
  2. “tes- tosterone” (should be “testosterone”)
  3. “rec- ommendations” (should be “recommendations”)
  4. “expo- sure” (should be “exposure”)
  5. “ad- equate” (should be “adequate”)
  6. “in- vestigation” (should be “investigation”)

Removing these unintentional hyphens will improve readability.

Final remarks:

1.     As a reviewer, I recommend that the authors revise the “Conclusions” to emphasize their findings' practical and clinical implications without restating specific results. Please focus on how 4000 IU/day vitamin D₃ supplementation reduces deficiency and supports sufficient vitamin D levels in elite swimmers. Highlight the role of individualized dosing and suggest more frequent (especially winter) testing to optimize and avoid excessive levels. This succinct approach will give clear, actionable guidance for practitioners and avoid duplication of the detailed data reported earlier.

2.     I recommend including a “Limitations section” to strengthen the manuscript's scientific rigor. Key aspects to address include the lack of a control group, which restricts causal interpretation, and the seasonal timing of measurements, which may not fully capture fluctuations in vitamin D levels. Additionally, the study's focus on elite British swimmers’ limits generalizability, and potential confounders, such as dietary intake and sun exposure, were not controlled. Clarifying these factors will provide a more balanced interpretation of the findings and ensure the manuscript aligns with standard reporting practices.

Author Response

(The authors gave the same response as above.)

Reviewer 3 Report

Comments and Suggestions for Authors

Evaluation of manuscript nutrients-3442531

This is an important manuscript about vitamin D supplementation in swimmers. I will present below my considerations about the text, point-by-point.

I would like to suggest changes to the title, I believe that a more direct title might attract more reader interest. I suggest something like: Vitamin D supplementation improves status in winter swimmers. Please review.

The Abstract is adequately structured. The introduction is clearly concise and has the appropriate elements, except the hypothesis, please add the hypothesis at the end of the introduction.

The methods are also well described, but I ask the authors to create a subhead at the beginning of the methods: 2.1. Experimental approach, where authors must present a general description of the experimental protocol and present the ethical aspects.

Enter the number of hours trained by the athletes per week and the competitive level of the athletes into the methods.

Why did the authors choose to supplement athletes who had between 81-125 nmol∙L− 1 considering that it is a range in which they consider that athletes already have sufficient levels of vitamin D.

Table 2 does not need a legend, the categories can be described alongside the concentrations.

Given the effects of vitamin D on immunity, has there been any monitoring of a lower rate of infection in athletes? It would be interesting if the study tried to only measure biochemical indicators and there were also practical applications of the effect of supplementation. Furthermore, it is important that authors give voice to authors who argue against the need for vitamin D supplementation.

https://link.springer.com/article/10.1007/s40279-017-0841-9

https://link.springer.com/article/10.1007/s40279-017-0841-9

Therefore, I think it is important that the authors also analyze whether there are symptoms of diseases caused by insufficient vitamin D. After all, if the athletes did not have any health problems, would it really be necessary to supplement vitamin D? I would like to suggest that the authors include reflections on what I suggested above in the discussion.

Author Response

(The authors gave the same response as above.)

Round 2

Reviewer 1 Report

Comments and Suggestions for Authors

Authors have carried out changes and have made an important effort to improve the manuscript within the possibilities taking into account the flaws in the study design. Unfortunately, I think many risks of bias were not considered. Thus, I cannot recommend the acceptance of this article.

Author Response

Author response: We are sorry to hear the reviewer has made this judgment on the manuscript. However, we are unable to adjust if the many risks of bias are not described. We have many multiple adjustments from the first revision, and we are unsure what risk of bias. We have gone back to the initial reviewer comments and made some further changes to the manuscript.

Reviewer 2 Report

Comments and Suggestions for Authors

I did not find the answer to ethical considerations: "While the study mentions ethical approval, there is limited detail on whether participants were fully informed of potential risks associated with high-dose vitamin D supplementation." It is necessary to acknowledge this concern in the methodology or ethical section.

  1.  

Author Response

Author response: We have now added the additional information in respect of the vitamin D. This was in our participant information sheet for the study as per normal research procedures. The revision can be seen on line 85.

Reviewer 3 Report

Comments and Suggestions for Authors

The previous version of the text was already very good, I suggested small changes. I am in favor of publishing the text, however, the authors should add another limitation, this one regarding the supplementation protocol, for which they were not responsible and instead followed the recommendations of British Swimming’s Chief Medical Officer.

Author Response

Author response: We have included this on line 125-127. This is what made the study a quasi-style experiment as we didn’t control all interventions.
